# Exploring the Cellular and Molecular Mechanism of Discoidin Domain Receptors (DDR1 and DDR2) in Bone Formation, Regeneration, and Its Associated Disease Conditions

**DOI:** 10.3390/ijms241914895

**Published:** 2023-10-04

**Authors:** Arokia Vijaya Anand Mariadoss, Chau-Zen Wang

**Affiliations:** 1Orthopaedic Research Center, Kaohsiung Medical University, Kaohsiung 80708, Taiwan; mavijaibt@gmail.com; 2Graduate Institute of Medicine, College of Medicine, Kaohsiung Medical University, Kaohsiung 80708, Taiwan; 3Regeneration Medicine and Cell Therapy Research Center, Kaohsiung Medical University, Kaohsiung 80708, Taiwan; 4Department of Physiology, College of Medicine, Kaohsiung Medical University, Kaohsiung 80708, Taiwan; 5Department of Medical Research, Kaohsiung Medical University Hospital, Kaohsiung 80708, Taiwan; 6College of Professional Studies, National Pingtung University of Science and Technology, Pingtung 912301, Taiwan

**Keywords:** DDR1, DDR2, cartilage, bone regeneration, tyrosine kinase, osteoarthritis

## Abstract

The tyrosine kinase family receptor of discoidin domain receptors (DDR1 and DDR2) is known to be activated by extracellular matrix collagen catalytic binding protein receptors. They play a remarkable role in cell proliferation, differentiation, migration, and cell survival. DDR1 of the DDR family regulates matrix-metalloproteinase, which causes extracellular matrix (ECM) remodeling and reconstruction during unbalanced homeostasis. Collagenous-rich DDR1 triggers the ECM of cartilage to regenerate the cartilage tissue in osteoarthritis (OA) and temporomandibular disorder (TMD). Moreover, DDR2 is prominently present in the fibroblasts, smooth muscle cells, myofibroblasts, and chondrocytes. It is crucial in generating and breaking collagen vital cellular activities like proliferation, differentiation, and adhesion mechanisms. However, the deficiency of DDR1 rather than DDR2 was detrimental in cases of OA and TMDs. DDR1 stimulated the ECM cartilage and improved bone regeneration. Based on the above information, we made an effort to outline the advancement of the utmost promising DDR1 and DDR2 regulation in bone and cartilage, also summarizing their structural, biological activity, and selectivity.

## 1. Introduction

Tyrosine kinase, a single transmembrane-bound cytoplasmic catalytic domain, is found on the polypeptide receptor cover in a group of significant cells. Tyrosine kinases are classified as either receptor or non-receptor tyrosine kinases. There are 90 known tyrosine kinases, 58 of which are receptor tyrosine kinases with 20 subfamilies, and the remaining 30 are non-receptor tyrosine kinases (NRTK) with 10 subfamilies [1]. Compared to NRTK, RTK is well known for its ability to regulate and stimulate growth factors and perform numerous signaling functions. RTK is activated by attaching the ligand to its receptor and dimerizing the receptor with a domain that forms the tyrosine kinase domain (TKD), after which additional steps are taken [2]. Receptor tyrosine kinases (RTK) bind to various ligands, including collagen, and regulate cell differentiation and proliferation to cellular morphogenesis, cell adhesion, migration, and incursion [3]. Researchers have discovered two self-promoting collagen receptor groups, one unique and containing discoidin homology kinase, even though it was classified as DDR1 and DDR2 among various DDRs from RTK until 1997 (Figure 1). DDRs have also been demonstrated to regulate immune activities. DDR1 is articulate in stimulated peripheral blood mononuclear cells and activated T cells, and it can regulate monocytic and T cell motility in three-dimensional (3D) collagen matrices [4]. DDR2, expressed in circulating human neutrophils, was discovered to play a similar role. In neutrophils, DDR2 favors the migration in 3D collagen matrices and enhances chemotaxis by activating MMP-8 and producing chemotactic collagen peptides. This protein is expressed in various cell types and may also play a role in wound repair, tumor cell proliferation, and invasiveness. The DDR2 mutations cause the short limb-hand-type of spondylometaepiphyseal dysplasia [5].

Furthermore, DDR1 and DDR2 regulate cellular morphology via receptor-mediated signals via the RTK kinase family. This DDR is also an extracellular region protein, with nearly 155 amino acids considered in the discoidin homology domain [6]. Discoidin domain receptors (DDRs) are classified from the tyrosine kinase subfamily based on their structure and physiological function [7]. DDRs have been related to various human illnesses, including fibrotic disorders of various organs, arthritis, atherosclerosis, and several forms of cancer. Some of the essential clinical significance of DDR associated with different types of diseases are listed in Table 1. So far, many different types of receptors have been discovered. On the other hand, several have a more extended history, as do their detection and evaluation. 

## 2. Discoidin in Various Genomes 

In particular, receptor tyrosine kinases are a novel subfamily that differs from another member of the large RTK family, and the homology domain, discoidin, was discovered throughout the cell aggregation process in the slime mold, Dictyostelium discoideum [36]. Following that, this unique RTK subfamily was cloned, and various research laboratories produced different cDNA copies. After cDNA cloning, they discovered that two distinct genes had been renamed DDR1 (previously Cak, DDR, RTK6, Ptk-3, or NTRK4, MCK-10, TrkE,) and DDR2 (previously called CCK-2, TKT, Ty-ro-10) [37]. Dictyostelium discoideum protein homology results in DDRs performing multiple functions rather than just cell adhesion, even though we knew nothing about DDR ligands at the time of discovery [38]. Later discovery of collagen identified as a physiological ligand for orphan receptors like DDRs will help to see the responsibility in cellular mechanisms [39]. 

The alternative splicing of DDR1 favors the three isoforms: a, b, and c. Exon 11 in the DDR1 juxta membrane region codes for 37 different amino acids for transcription, whereas it is absent in the DDR1a sub-subfamily. However, it was found in DDR1b from another sub-subfamily. Isoform-C would form if another set of six amino acids (S-F-S-L-F-F-S) were added to the kinase domain. When DDR1 and DDR2 are over-expressed in cells, they show 125 and 130 kDa of glycosylated proteins, respectively [40]. According to reports, the DDR1b isoform improves rat postnatal development compared to the DDR1a isoform [41]. In an in vivo study, a rat treated with tunicamycin resulted in DDR1a and DDR1b with 102 and 106 kDa, indicating that shorter isoforms were glycosylated at a higher percentage than longer ones [42]. Another study found that DDR1a is synthesized at 63 kDa, then anchored to the protease membrane before being converted to 54 kDa and solubilized, forming furn-like proteases [43]. 

The DDR2 gene in Saccharomyces cerevisiae is a multi-stress-response gene transcriptionally activated by environmental, xenobiotic, or physiological factors. The DDR2 gene encodes a short hydrophobic 61 amino acid polypeptide found on chromosome XV near the SPE2 locus [44]. Labrador et al. (2001) discovered DDR2 expression along chondrocyte columns in the proliferative zone of the growth plate using in situ hybridization on 1-week-old mice. Though scattered, DDR2 mRNA was also found in calcified cartilage in the cartilage bone interface and on the trabecular bone surface [45]. DDR2 protein was found in the majority of mouse tissues studied. Excessive amounts of phosphorylated DDR2 were noticed in the lung, ovary, and skin, but these levels did not correspond with DDR2 protein levels. Both DDRs can interact with multiple fibrillary collagens; however, DDR2 requires network-forming collagen IV to be started [45]. Despite this, the interaction of these DDR2 receptors with collagen results in auto-phosphorylation, the first step in transmembrane signaling. Leitinger and colleagues (2006) isolated hypertrophic chondrocytes from chicken embryos that efficiently expressed DDR2 [46].

## 3. Expression of DDRs and Related Proteins 

The Expression of DDRs is studied using Northern blot and in situ hybridization in most human and mouse hepatocytes. The DDR1 expression is restricted in mouse development in the Langerhans islets. It can also be used as a primary marker for neuroectodermal growth and development in mice [47]. The northern blot reveals that DDR2 is expressed primarily in the skeletal muscle and heart tissue, with less expression in the brain, lung, and kidney connective tissues [48]. More than five studies suspected and confirmed DDR1 overexpression in human tumors. Aside from its expression in tumor cells, it was discovered that DDRs or any one subfamily of DDR cellular signaling allowed for a targeted deletion mutation in embryonic stem cells [39]. A recent study used knock-out mice during embryogenesis and post-adult-stage development to claim the role of DDR1 and DDR2 [49].

According to gene bank databases, researchers identified three homology domain proteins for the Caenorhabditis brigade and C. elegans genomes, which activated DDR1 and DDR2 cellular signaling. At the same time, the above three proteins resemble DDRs and share characteristics such as a long juxta membrane stretch, the amino terminal of the discoidin domain, and a catalytic tyrosine kinase protein in the c-terminus. However, they could not identify any significant role for the three proteins mentioned above, discovered by chance during worm genome research [50,51]. Even though collagen can induce these receptors, a recent study demonstrated that RTKs could perform the same function as DDRs in nematodes and mammals, namely the action of epinephrine in the nervous system during an emergency. 

Another surprising finding is that DDR1 of the tyrosine kinase family is closely associated with the marine sponge, Geodia cydonium, genome. In contrast, 59%–61% of the catalytic pattern in DDR1 and geodia tyrosine kinase are nearly identical to RTK. Furthermore, the ancestor pattern for Geodia and DDR1 appears to have been developed 600 million years ago during primary evolution and multicellular organism development; in addition to NTRK, the discoidin domain is found in two other mammalian proteins, neuropilins and neurexins. The name implies these mammalian genomes played a role in nervous system development [52,53].

The previously identified genome of Xenopus laevis belongs to the neuropilins and is also known as the A5 antigen, acting as a receptor for semaphorins. It is a glycoprotein with some isoforms found in growth factors. Tandem repeats are found in neuropilins and their DS domains, flanked by other fields in cell adhesion proteins [54]. Another intriguing discovery is that the neuropilin/A5 antigen domain is followed by a nearly 80 amino acid stretch, which serves as the corresponding extracellular domain region for DDR2 [55]; furthermore, the collapse of semaphorin growth was influenced by sensory and motor neuronal interactions. As a result, DDR1 of the DDRs is expressed in various forms and promotes distinct protein synthesis and growth during cellular development are produced by DS domain progenitors, improving cell aggregation and signaling to convert one-cell organisms, such as amoebas, to multicellular organisms. Several studies have shown that DS domains can interact with mammalian proteins such as lectins [56,57,58]. Most reviews have gone into great detail about the role of DDRs and their associated downstream signaling in various human diseases. They have concentrated on DDR structural ligand recognition and activation ligand binding specificity. When researchers decided to identify the ligand that carries DDR through two different individual laboratories, they discovered that collagen is the ligand that activates DDRs. The main question is how ligand binding becomes receptor activation. Initially, it was thought that RTK activation causes the ligand-receptor dimer to be converted into systolic tyrosine kinase, which promotes transautophosphorylation. Dwarfism, SMED, and short limb-hand type are rare symptoms caused by mutations in the human DDR2 gene [59]. Three out of four discovered missense mutations cause the reversal of mutated proteins in the ER via MAP kinase signaling controlled by DDR receptors. The E113K mutant gene is generally expressed on cell surfaces; however, it does not bind with collagen X, resulting in the eventual shortening of long bones. The Arg105-Glu13 salt bridge distinguishes the OH-pro in the GVMGFO motif, as demonstrated by artificial crystal DDR2 with Ds collagen mixtures [60].

## 4. Collagen: The Prime Activator of DDRs Signaling

Collagen is not only a DDR1 and DDR2 ligand. It also acts on DDRs to release signals that promote matrix protein and genome physiological growth. Currently, 19 different types of collagen are as common as the sheet-like fiber collagens 1, 2, 3, and 4. The remaining collagens are rarely expressed and play a minor role in cell and tissue size regulation [61]. Inadvertently, collagen-coding genes are reduced or deleted, resulting in a link between collagen failure and severe disease in the skeleton, skin, and tendons. Even though several studies have revealed that all 19 types of collagen could activate DDR1, DDR2 is started by specific types of febrile collagen, and the activation is prolonged, it appears to be ongoing more than 18 h after collagen was added. Additionally, no significant down-regulation was observed after four days, and activation in DDR2 is maintained by febrile collagen. The innate and actual configurations of collagen are required to activate DDR1 and DDR2, whereas glycosylation alone is sufficient to enhance DDR2 [62].

Collagens 1 and 2 have a similar magnitude and increased affinity unless coupled with collagen IV. At the same time, both DDRs and their interactions with collagen are specific triple helices [63]. DDR2 could not recognize collagen IV and its associated basement membrane in a study on collagen binding receptors, and two decades of collective information on them was observed as fibrillary collagen I, II, III, and IV [64]. Meanwhile, a previous study reported that the physical interactions of collagen X and its cell collagen X have significant reactions with DDR2. DDR1 weakly connects with collagen X due to its low affinity for the receptor. DDR2 and its interaction with cartilage-specific collagen II yielded the same results [65].

Collagen X and its expression are limited for long bone growth, according to Labrador et al. (2001), whereas DDR2 and its broad face in the body are associated with hypertrophic chondrocyte cells. Their findings suggested that DDR2 protein is synthesized by collagen X from hypertrophic chondrocytes expressing mRNA. As a result, DDR2 advocates for collagen X in the growth plate, citing its strong relationship and the fact that it is a physiological receptor for X [45]. Another in vivo study found that DDR2 expression in proliferating murine chondrocytes and DDR2 deletion or removal in the same mouse results in affected bone growth or reduced chondrocyte proliferation, leading to a physiological growth defect in the mouse. In any case, RT-PCR and immunohistochemistry failed to identify the DDR2 protein and its accumulation in the murine growth plate. According to their findings, the expression of DDR2 at the junction of hypertrophic proliferative colonies results in DDR2’s role in cell maturation and proliferation [66]. DDR2 protein is abundant in all articular cartilages, while the collagen X expression is constrained to the deep zone of pericellular chondrocyte hypertrophic regions. However, the reason DDR2 is not found on the growth plate during chondrocyte proliferation remains unknown [67]. However, DDR2 protein and mRNA expression confirm that it plays a prominent role in bone growth. Leitinger et al. (2008) discovered that the DDR2 domain eventually has the binding sites designed for fibrillary collagens I and II. On the other hand, Collagen I and its binding site are represented by three spatially adjacent surface loops of the DDR2 [46].

Meanwhile, the surprising discovery is that the DDR2 discoid domain does not recognize collagen X, implying that it requires something else to be recognized in DDR2. It also confirms that the non-fibrillar collagen-binding mechanism in DDR2 is more diverse than the fibrillary collagen binding mechanism [68]. Furthermore, based on existing studies, which have thoroughly investigated the critical difference in DDR2 significance between fibrillary and non-fibrillar collagens, collagen X binding with DDR2 differs from collagen X binding, with the 21 integrins of triple helical dimension required for DDR2 protein binding [69]. Most studies have discovered that collagen X is necessary for growth plate development, but cannot exist without DDR2 binding. Based on biological growth changes, one study found that the tyrosine kinase domain activates DDR2 and promotes cell proliferation in the growth plate. However, the mechanism by which the collagen molecule docks the DDR1 and DDR2 through its epitopes remains unknown. Synthetic collagen composed of ten repeated collagens was attempted in order to form a triple helix, but it was insufficient to stimulate DDRs [70]. The significance of DDR signaling remains unknown. Another study discovered that MMP-1, a significant collagen fibrillary upregulating enzyme, improved the human skin fibroblasts with DDR2 [71]. Collagen is a ligand for both DDRs and integrins. The fibroblast integrin is activated by fibroblasts, an important cytoskeleton growth initiator for collagen formation through the DDRs delivering the signaling process [72]. Furthermore, it has also been reported that 3D-cultured collagen gels raise the question of whether mechanical or external forces induce DDR phosphorylation. Lund et al. (2009) documented the interaction of DDR1 with three-dimensional type I collagen using a 3D culture of human mesenchymal stem cells. The dynamic cell shape changes and ECM microstructure tuning cause DDR1 and two-dimensional osteogenesis pathways to interact and modify their functions [73]. Meanwhile, DDR1 signals activate β-integrin, which is required for fibroblast and epithelial cell development [74].

## 5. Molecular Signaling of DDRs in Bone and Cartilage 

In bone, DDR1 signaling is crucial for regulating osteoblast (bone-forming cell) function and remodeling. When activated, DDR1 initiates a cascade of intracellular signaling events that promote osteoblast differentiation and activity [75]. It stimulates the expression of bone matrix-synthesis-associated genes, such as collagen type I and osteocalcin, leading to the deposition of new bone tissue. Additionally, DDR1 signaling regulates osteoblast migration, facilitating their recruitment to sites of bone remodeling [10].

The DDR1 signaling pathway significantly stimulates bone regeneration. As stated previously, DDR1 is a receptor tyrosine kinase activated by collagen, an essential component of the extracellular matrix of bone tissue. When collagen binds to DDR1, intracellular signaling events modulate bone cell activity and promote bone remodeling and regeneration. The initial step in the DDR1 signaling pathway is the binding of collagen. DDR1 is activated when collagen molecules bind to its discoidin domain in the extracellular matrix. This binding induces receptor dimerization, which assembles the intracellular kinase domains of DDR1 [76]. DDR1 dimerization causes the autophosphorylation of specific tyrosine residues in intracellular kinase domains, which is the next step in the signaling pathway. This autophosphorylation enhances the kinase activity of DDR1 [77]. Diverse signaling molecules, such as adaptor proteins and kinases, bind at activated DDR1. The phosphotyrosine-binding (PTB) domains or Src homology 2 (SH2), along with other phosphotyrosine recognition modules have the potential to interact with the autophosphorylation event resulting from the dimerization process within the DDR1 signaling pathway [78]. This autophosphorylation enhances the activity of the DDR1 kinase. The p85 subunits of phosphatidylinositol-4,5-bisphosphate 3-kinase (PI3K), SHP-2, and ShcA attach to the internal juxtamembrane region and the tyrosine kinase domain of DDR1. These signal molecules bind to the phosphorylated tyrosine residues on DDR1 to start the downstream signaling cascades [79]. Ras/ERK MAPK and PI3K/Akt cascades appear to be activated in response to collagen stimulation of DDR1. The DDR1 activation favors the regulation of several downstream signaling pathways, such as the JAK/STAT, PI3K/Akt, and MAPK/ERK pathways [29]. These pathways control gene expression, cellular growth, differentiation, and survival. These signaling molecules bind to the phosphorylated tyrosine residues on DDR1 to initiate downstream signaling cascades [80,81]. Additionally, it encourages the differentiation of osteoblasts. To promote bone formation, DDR1 signaling affects osteoblast differentiation and function. Moreover, the differentiated osteoblasts have been found to promote osteogenic transcription factors, including Runx2 and Osterix. During osteogenesis, the knockdown of DDR1 in osteoblasts decreased the ALP activity, mineralization, phosphorylated p38, and protein levels of Runx2, BMP2, Col-I, ALP, and OC in OKOΔDDR1 mice. Overexpression and DDR1 knockdown in osteoblasts showed that DDR1 regulates the p38 phosphorylation mechanism to regulate Runx2 and the downstream osteogenesis markers during osteogenesis. DDR1 also increases the production of extracellular matrix proteins like osteocalcin and collagen to form bone matrix [10].

Osteoclast activity is the crucial physiological process for bone remodeling, and it is additionally impacted by DDR1 signaling. Osteoprotegerin (OPG) is encouraged to express itself. By suppressing the expression of RANKL, a protein that promotes osteoclast development and activity, OPG prevents osteoclastogenesis and inhibits osteoclastogenesis [82]. Zhang et al. (2020) revealed the mechanism of DDR2 in chondrogenic and osteogenic differentiation. They found that the induction enhanced DDR2 activation in preosteoblastic cells without altering DDR2 expression. In the differentiated conditions, the downregulation of endogenous DDR2 by specific shRNA inhibited the osteogenic differentiation and osteoblastic marker gene expression [83].

DDR1 signaling promotes the production of factors that regulate osteoclast activity, such as RANKL (Receptor Activator of Nuclear Factor Kappa-B Ligand), which stimulates osteoclast formation and bone resorption. This delicate equilibrium between bone formation and resorption is crucial for maintaining bone integrity and homeostasis [9]. DDR’s protein tyrosine kinase receptors regulate bone formation after binding to collagen. DDR1 originates in epithelial cells, whereas DDR2 originates in mesenchymal cells [84]. SMED (spondylo-meta-epiphyseal dysplasia) is a type of limb-hand dysplasia characterized by short limbs, short stature, and broad fingers with abnormal epiphyses and premature calcification [85]. Even though the absence of DDR2 signaling calcification results in bone fractures or wide and shorthand fingers, Borochowitz reclassified it in 1993 as a type of congenital familial skeletal dysplasia with histopathological damage to pathological damage [86]. 

Meanwhile, hypertrophic chondrocytes are distinguished by the upregulation of proteolytic enzymes triggered by ECM degradation. As a result, the mechanisms underlying the differentiation of hypertrophic chondrocytes with ECM degradation enzymes are critical in developing effective treatments for OA. When tyrosine kinases (TKs) bind to collagen and become activated, DDRs interact with cell-collagen communications in normal and pathological conditions [87]. DDRs also regulate cell differentiation, migration, adhesion, and tumor metastasis, including arthritis [39]. Collagen IV, II, and III can activate DDRs. Depending on the receptor type, DDRs are activated by collagen IV, II, or X, which is activated by molecular signaling via PI3K/AKT [88]. OA is classified as pathogenic based on the affinity binding between collagen II and X and DDRs. Most in vitro research and investigations have shown that DDR2 and its role in OA cause joint injuries in tissue joints due to triggered collagen II cleavage, which results in DDR2 activation by TK autophosphorylation [89]. Another study discovered that DDR2 activation induced the over-expression of hypertrophic markers such as MMP-12, Alpl, and Col10a1 [90]. 

According to Borochowitz, the shortened limbs result from a missense mutation in the DDR2 gene, which fails in the ligand-binding function. Long bones become shorter in DDR2 knockout mice; the same phenomenon is seen in SMED patients. It is caused primarily by a lack of DDR signaling, which is also necessary for bone tissue development [80]. Another recent study discovered that DDR2 regulates bone markers and their expression by regulating osteogenic variation. DDR2 control of bone markers is most likely due to the activity of RUNX2, a major transcription factor that becomes entangled during bone differentiation [81]. RUNX2 action is modulated by phosphorylation in response to a signal from the extracellular regulated kinase (ER-K), which activates DDRs. Another study discovered that ER-K improves the upstream genome and that its activity is linked to DDRs through Shc and Src [91]. 

DDR1 signaling differentiates mesenchymal stem cells into chondrocytes responsible for cartilage production and maintenance. DDR1 activation increases the expression of significant transcription factors implicated in chondrogenesis, including Sox9 and Runx2 [92]. These transcription factors kickstart the chondrocyte differentiation. Moreover, DDR1 signaling is involved in chondrocyte differentiation and maturation. It regulates the expression of transcription factors, such as Sox9 and Runx2, which play critical roles in chondrogenesis and endochondral bone formation [93]. DDR1 also influences chondrocyte proliferation and survival, contributing to the overall maintenance of cartilage tissue via phosphorylation of p38. DDR1 signaling is a vital pathway in bone and cartilage, orchestrating cellular processes essential for tissue development, maintenance, and remodeling. It regulates the delicate balance between bone formation, resorption, and cartilage’s structural integrity and function [76]. Understanding DDR1 signaling and its complex interactions in bone and cartilage may have implications for developing therapeutic approaches targeting musculoskeletal disorders, such as osteoporosis and osteoarthritis.

## 6. DDRs Expression in Bone and Tissue

Human physiology or pathophysiology of balanced tissue homeostasis is achieved through the interaction of cells with their environment. As a result, the interaction occurs via bone tissue and a process known as remodeling. It maintains those physiologies by allowing adaptable bone molecules to form during damage or regeneration. Although the mechanisms of bone remodeling known as resorption and deposition have not been studied, the extracellular matrix is a network of bound macromolecules found in the multicellular organism of cells and connective tissues (ECM). It was discovered to be collagen, a protein that provides adhesive properties and is the most abundant ECM macromolecule [94].

DDRs are primarily expressed in osteoblasts, which are responsible for bone formation, and the osteoclasts mechanism favors the bone resorption mechanism in bone. Several researchers have documented the DDR1 and DDR2 expression in osteoblasts, and these receptors have a crucial role in osteoblast differentiation and function [95]. A downstream signaling event that controls osteoblast activity, such as cell migration, proliferation, and formation of extracellular matrix elements like type I collagen, is triggered when their ligands activate DDRs, which are collagen molecules in the extracellular matrix [39]. Similarly, DDR1 has been linked to osteoclast differentiation and bone resorption and is expressed by osteoclasts. In osteoclast precursors, activation of DDR1 triggers signaling pathways that encourage osteoclast activity and development. However, there is conflicting evidence about DDR2 expression in osteoclasts, with some research indicating it is present and others indicating it is not. The function of DDR2 in osteoclasts requires more investigation [96].

The critical concern is that the onset of pathogenesis and fundamental OA metabolism influences DDR-mediated matrix degeneration. On the other hand, DDR2’s effect on MMP-13 and its expression causes significant damage to bone formation and regeneration, leading to deformities [97]. Numerous researchers have acknowledged the importance of DDR-mediated MMP production in diseases if it is dysregulated. It is also necessary to understand physiological expression in homeostasis and its role in DDR-induced MMP activation or its manifestation in conditions. The loss of chondrocyte proliferation in DDR1 knockout mice results in shortened long bones, dwarfism, and a smaller snout [45], in line with previous studies involving chondrocyte-specific (a1(II) collagen, CreERT; Ddr1f/f mice) using CKO mice and DDR1 knockout mice. The secondary ossification center’s delayed development and inhibited the growth plates in the rear limbs of the CKODdr1 mice. Moreover, DDR1 deletion in chondrocytes reduces the Ihh/Gli1/2/Col-X signaling, cell terminal differentiation, and apoptosis [93]. A recent study discovered an impulsive autosomal recessive mutation in mouse colony culture that results in DDR2 gene deletion. This DDR2-deficient mouse, also known as a “slie,” is a dwarf and sterile. Mice (DDR2slie/slie) with a non-functional DDR2 allele (Smallie) have multiple skeletal defects. When comparing wild-type, DDR2wt/slie, and DDR2slie/slie mice, a gradual decrease in tibial trabecular BV/TV was observed. These alterations were accompanied by decreased trabecular number, thickness, and spacing in both males and females. Additionally, the researchers investigated the role of the DDR2 signaling mechanism in BMSCs cultured under osteogenic and adipogenic conditions. DDR2slie/slie cells exhibited defective osteoblast differentiation and enhanced adipogenesis, and the expression of RUNX2 and PPAR was regulated by MAPK-dependent phosphorylation [98,99].

A 6-day-old mouse study revealed that DDR1 chimeric fusion protein was expressed, as well as the presence of placental ALP. It left traces in the skeleton, skin, and UT when sacrificed. In addition, abnormal or overexpressed expression of both DDRs has been found in cancer cells [100]. DDR1 and its regulation are observed in atherosclerosis, myxomatosis, rheumatoid arthritis, and osteoarthritis. DDR2 has also been linked to the principle of atherosclerosis and lymphanmyomatosis [19]. Meanwhile, according to the researchers, infertility of sperm is due to spermatogenesis defects caused by DDR-controlled gonadal expression. Although it has not been reported in DDR2-/- mice, any additional deficiencies may play a role in the cause of infertility in slie mice [101]. DDRs are a critical bone growth-regulating factor that regulates several aspects of bone growth. Although DDRs regulate endochondral ossification and maturation, Runx2 is a crucial transcription factor in osteoblast and chondrocyte differentiation. However, it does not consider which collagen and its ligands were triggers for DDRs to manage endochondral proliferation and ossification [66]. An enzyme called lysyl oxidase catalyzes the cross-linking of collagen fibers, which modifies bone strength.

In contrast, as mentioned earlier, lysyl oxidase is secreted by DDR–collagen interaction in osteoblasts for bone strength [102]. Another study found that high DDR2 expression in C57BL/6 x DBA transgenic mice influences body size physics and that this is the only significantly different parameter from the other normal parameters listed. Transgenic mice are also longer and weigh less, resulting in a low BMI, according to CEPSCA guidelines. Transgenic mice appear to have high leptin production, resulting in low epididymal body fat. As a result, it is proposed that DDRs regulate fat metabolism, absorption, and storage by maintaining skeletogenesis and bone growth. However, more research is needed to confirm whether DDRs are directly linked in signaling to leptin production [103]. Numerous studies have found that both DDRs play an essential role in development, particularly DDR1 in organogenesis, which includes fibroblasts with diverse pedigrees, and DDR2 in bone growth, which provides for chondrocytes and osteoblasts [84]. Due to the small size characterization of knockout mice, Vogel et al. (2001) discovered deficiencies in DDR1. XX (females) exhibited various reproductive defects and a decreased blastocyst, resulting in a higher percentage of XX knockout mice becoming infertile. The most visible deficiency is abnormal mammary gland branching, which results in milk secretion failure [100].

DDRs were discovered in different tissues of the adult brain after growth. DDR1 mRNA is observed in nearly all tissues of mice and humans, even though DDR1 is in high concentrations in the human lung, brain, spleen, kidney, and placenta [104]. Meanwhile, DDR2 mRNA is found primarily in the skeletal and cardiac muscles, with less in the kidney and lungs. In addition, these DDRs are found and expressed in the human nervous system [105]. Aside from the discovery of DDRs in various tissues and organs, a discussion about their expression in tissues or systems and how they support human health began. DDR1 is primarily expressed in epithelial tissue, whereas DDR2 is found in connective tissues derived from the mesoderm in the embryo [64]. Few studies have found that DDRs can also be found in innate immune systems. However, no comprehensive or organized cellular expression of DDR proteins for various tissues has been performed.

In the meantime, DDRs play an essential role in embryonic development, but their role in adult tissue remains unknown. DDR1 function is necessary for the development of the mammary gland, though DDR2 is crucial in the growth of long bones. DDR1 activation in response to collagen binding during embryogenesis may favor the cells and tissue organization. DDR1 expression is also linked to maturation of the brain cells. It has been detected in neural precursor cells and neurons during nervous system development [106]. In wild-type mice, DDR1 is expressed at every stage of mammary development. In late-stage pregnant DDR1/mice, the mammary glands displayed a compressed alveolar form, with the fat pad packed with ducts [100]. DDR1−/− animals have abnormalities of cell-autonomous and localized to the mammary epithelium as shown by transplantation experiment [107]. Besides, the DDR2 expression was identified in mouse and rat hearts, and it was principally expressed in the form of cardiac fibroblast [108], whereas DDR2 expression was placed on the epicardial surface of the heart [109]. Some studies have implicated DDR2 in cardiac fibrosis, which can affect heart function during development and adulthood.

Discoidin Domain Receptors (DDR1/2) are the members of Receptor Tyrosine Kinases (RTKs). There are two isoforms DDR1 and DDR2. The phosphorylation of the intracellular tyrosine domain decides the faith of the signaling cascade. Collagen an extracellular matrix structural protein found in connective tissue is the ligand for the activation of DDR1/2 approximately 28 types of collagens are identified but only a few are involved in activation. DDR1/2 have a crucial role in bone and cartilage development. The interactive pathways of DDR1/2 are MAPK, PI3K, JAK/STAT, and Rho-GTPase. The signal transduction induced by DDR1/2 also activates ERK1/2, Akt, cytokine signaling, and RhoA signaling pathway, respectively, and play a role in cytoskeletal dynamics, cell proliferation, differentiation, survival, adhesion, migration, cell metabolism, cytokine signaling, and immune responses. The possible molecular signaling pathways of DDR1 and DDR2 in association with bone and cartilage are shown in Figure 2.

## 7. DDRs as a Potential Therapeutic against OA and RA

In the previous two decades, Hou et al. (2012), Franco et al. (2010), and other researchers (2006) identified how the regulation of DDRs, particularly DDR1, fails in expression and leads to disease, most notably bone and brain disorders. According to reports, the mechanism by which DDRs regulate cell adhesion and migration behavior has yet to be fully discovered [18,110,111]. DDRs have been linked to multiple pathological conditions, including osteoarthritis (OA), rheumatoid arthritis (RA), and cancer, and they are being explored as potential therapeutic targets [112]. Osteoarthritis (OA) is a degenerative joint disease depicted by the breakdown of cartilage and the under-lying bone. DDRs have been found to play a role in cartilage homeostasis and matrix remodeling, critical processes involved in OA progression [113]. Prior studies have reported that the increased DDR expression is associated with cartilage degradation in OA. Additionally, some studies proposed that the signaling of DDR is a potential therapeutic approach to prevent cartilage breakdown and slow the progression of OA. A study by Manning et al. (2006) offers experimental evidence that DDR2 may be an attractive target for developing disease-modifying OA medications. DDR2 was conditionally eliminated from the articular cartilage of Aggrecan-CreERT2 mice. The progressive process of articular cartilage deterioration was significantly slowed in the knee joints of DDR2-deficient mice compared to their control mice. Damage to articular cartilage in the knee joints of mice was related to elevated expression levels of DDR2 and matrix metalloproteinase. These findings imply that DDR2 may be an appropriate target for developing disease-modifying OA medicines [114]. Sunk et al. (2007) investigate the connection between upregulated DDR2 expression and cartilage degeneration in OA. This investigation used cartilage tissue samples from 16 human knee joints to examine the expression of MMP-13, DDR2, and MMP-derived type II collagen fragments. The immunohistochemistry study showed that the expression of DDR2 in human articular cartilage increased with increasing tissue injury. At the same time, cartilage degradation with higher DDR2 expression led to a more significant release of MMP-13 and type II collagen breakdown products [14]. A similar pattern of results showed the up-regulation of DDR2 in chondrocytes in the articular cartilage of knee joints in mice exhibiting OA due to a mutation in type XI collagen. The activation of DDR2 initiates a cascade of cellular processes that ultimately lead to the release, synthesis, and activation of matrix-degrading proteinases, specifically MMP-13, which leads to the development of OA [115]. Our recent findings also revealed that the inhibition of DDR1 minimizes osteoarthritis through the autophagy mechanism. The results indicate that the intra-articular injection of 7rh molecules effectively diminished the cartilage deterioration in C57BL/6 mice with osteoarthritis produced by anterior cruciate ligament transection. The suppression of DDR1 has been shown to reduce the hypertrophic differentiation of chondrocytes and chondrocytes apoptosis in OA. Additionally, it can restore the reduced autophagy function caused by OA [11]. 

OA is a potentially fatal morbidity and dominant joint disorder affecting more than 60% of people over 40. It is due to poor calcification or uric acid deposition in joints and the release of molecular signaling receptors [116]. Degeneration of articular cartilages associated with failure in subchondra. Bone remodeling regulation leads to inflammation in the synovial adjacent and, ultimately, loss of joint movements and pain [117]. As OA’s primary cause and risk factors have yet to be determined, no effective drug or preventative measure is available. According to the data from Pub-Med and arthritis journals, articular chondrocytes undergo phenotypic changes and accumulate fluid, resulting in OA. This also appears to be an endochondral ossification issue, a naturally occurring procedure in long bone elongation growth plates from the neonatal period to puberty [118].

Meanwhile, cartilage is synthesized in early embryonic stages on endochondral bone, which is differentiated as hypertrophic cartilage, and eventually enters apoptosis or is distinguished as osteoblast, resulting in bone replacement [119]. Conversely, rheumatoid arthritis (RA) is an autoimmune condition that causes joint damage and chronic inflammation while the host immune cells attack the healthy cells. DDRs are thought to play a vital role in controlling inflammatory and immunological reactions in these clinical conditions. The synovial tissue, the source of inflammation in the joints, has been found to express DDRs more frequently in RA [120]. Mu and colleagues discovered that inhibiting DDR2 lowers inflammation and joint destruction via the H19-miR-103a-IL-15/Dkk-1 axis, where DDR2 plays a stimulatory function in the development of RA. DDRs may help control the inflammatory response and stop the damaging effects on the joints. POSTN enhances the collagen and proteoglycan degradation of chondrocytes [121]. In line with previous studies, POSTN influences the proteoglycan collagen degradation in cartilage through DDR1. POSTN-induced MMP-13 expression was inhibited in mouse chondrocytes by genetic or experimental inhibition of DDR1. These findings revealed that POSTN signals via DDR1 are systematically implicated in OA’s pathogenesis. Specific DDR1 inhibitors may offer therapeutic options for treating OA. Liu et al. (2023) conditionally deleted the DDR2 in myeloid lineage cells to generate cKO mice to investigate the role of DDR2 in myeloid lineage cells. They found that cKO mice exhibited more severe inflammation in collagen-antibody-induced arthritis (CAIA) and high-fat diet (HFD)-induced obesity, indicating the protective role of DDR2 against inflammation. Mechanistically, DDR2 promotes macrophage repolarization are systematically implicated in OA’s pathogenesis from the M1 to M2 phenotype and protects the systemic inflammation [122]. Ge et al. (2008) discovered that DDR2 is preferentially expressed and activated in the articular zone of TMJs but not knee joints using DDR2 LacZ-tagged mice. The absence of DDR2 results in abnormalities in chondrocyte maturation and mineralization, according to research on primary cultures of TMJ articular chondrocytes from wild-type and Ddr2slie/slie mice. These investigations show that DDR2 functions are localized to the fibrocartilage of the TMJ and are not present in the hyaline cartilage of the knee and that DDR2 is required for appropriate condyle formation and homeostasis in the TMJ [123]. TMJ osteoarthritis could develop as a result of gene editing, which includes overexpressing the genes of transforming growth factor-1 (TGF-1), short stature homeobox 2 (SHOX), and β-catenin, as well as knockout or inhibit the DDR1, fibroblast growth factor receptor (FGFR3), and small mother in opposition to decapentaplegic 3 (SMAD3) [124]. After experiments on mice and mouse models, DDRs have been used to treat various chronic, long-term human diseases. DDRs play a positive role in pathology, and DDR inhibitors have been identified as a promising therapeutic method when treating specific conditions with limited options [7,39]. As their use by scientists and researchers grows daily, DDRs are now regarded as a bulleting target in drug development. Nilotinib, imatinib, and dasatinib are the protein kinase inhibitors that target the tyrosine kinase activity of the Breakpoint Cluster Region-Abelson kinase (BCR-ABL). These inhibitors are also utilized to treat chronic myelogenous leukemia [125]. Nilotinib is a small-molecule kinase inhibitor that targets the leukaemia-enhancing cluster region Abelson kinase protein. These drugs, however, inhibit DDRs and other potentially active kinase receptors [126]. VU6015929, a powerful DDR1 kinase inhibitor with improved physiochemical, DMPK, and kinome profiles, favors the development of the DDR1 inhibitor’s efficacy, as proven by the in slico model [127]. DDR1-IN-2 is another powerful DDR1 inhibitor that inhibits several other kinase targets. DDR1-IN-1 binds to DDR1 in the ‘DFG-out’ conformity and suppresses DDR1 autophosphorylation in cells at sub-micromolar concentrations with high selectivity as evaluated by the KinomeScan technology against a panel of 451 kinases [128]. The preclinical study also revealed that 7rh ((3-(−2-(pyrazolo [1,5-a] pyrimidin-6-yl)-ethynyl) benzamides) inhibited the DDR1-expression. It has an IC50 value of 6.8 nM for suppressing DDR1 enzymatic activity but is substantially less effective in overwhelming other kinase activities such as Bcr-Abl, DDR2, and c-Kit [129]. As a result, there are two types of subfamily receptors: DDR1 kinase inhibitors and DDR2 kinases. Even though a recent study found two groups described as optimistic, DDR1 kinase inhibitors and DDR2, DDR2 was ineffective against common antibiotics such as actinomycin-D [130].

According to studies and their findings, DDR2 is a critical receptor molecule in the development and progression of tumor migration, osteoarthritis, and bone development. According to a survey, overexpression of DDR2 with partial Function in RA patients increases RA synovial fluid while promoting cartilage-degrading matrix metalloproteases (MMP-1, MMP-2, and MMP-13) [131]. Meanwhile, the overall process of DDR2 depends on the pathophysiologic involvement of RA FLS-influenced DDR2-induced MMP and MMP subclass secretions [132]. RA is a chronic systemic disorder characterized by joint inflammation that leads to bone function loss, a physiology defect. Excessive inflammation leads to invasion because the synovial layer lining is a crucial feature in RA. DDR2 is a key promoter of collagen and its associated actions in ECM remodeling, fibroblast migration and differentiation, and neovessel formation. According to some studies, over-expression of DDR2 in RA synovial tissue causes cartilage and bone devastation in OA and RA, which MMPs regulate [133,134,135]. Another research finding stated that communication between DDR1 and collagen II in-duces the expression of MMP-13, which damages the RA cartilage. Finally, it is unclear whether DDR2 and its role in regulating bone and cartilage influenced the inflammatory response and how bone damage manifested in RA.

A study shows that it reduced the severity of OA in mice lacking DDR2 [136]. In addition, recent research has shown that inhibiting DDR1 in OA mice results in lower MMP13 and Col1a1 expression levels. Although these findings confirm and elucidate the role of DDR1 and DDR2 in stimulating chondrocyte hypertrophy [137]. There has been a surge of interest in the molecular signaling of TK and its pathophysiology in various disorders associated with receptors that play an essential role in controlling those conditions. It contains the receptors FGFR-1, DDR1, DDR2, EGFR, FAK, TrkA, and Fyn, a few of which have been identified as inducers of chondrocyte hyper-trophy and articular cartilage. As a result, the receptors mentioned above may impair the functions of TKs and can be considered a potential treatment for OA [89]. Meanwhile, articular cartilage and its homeostasis, regulated by TKs-regulated signaling receptors (particularly DDR2), may be dysregulated, leading to chondrocyte hypertrophy [138]. Although some repositioning drugs can be used as an alternative medicine at a low cost, they are best suited for developing countries.

Osteoarthritis (OA) is a set of disorders that can cause symptoms and signs in the joints, also known as chronic degenerative disorder. The signs and symptoms of OA are connected with a loss of integrity in the articular cartilage and changes in the underlying bone and joint margins [139]. However, the molecular mechanism underlying OA has yet to be discovered. Our recent study investigated the DDR1 inhibitor of 7 rh roles in OA prevention and autophagy mechanisms. According to our findings, the IA injection of 7 rh increased the OA joint functionality and decreased the OARSI score. Furthermore, we uncovered that 7 rh inhibits the production of MMP13, Col X, and IHH, which can reduce chondrocyte hypertrophic differentiation [11]. 

A high incidence of OA in the temporomandibular joint (TMJ) was seen in DDR1 knockout mice. The decrease in DDR1 expression significantly impacts the pathophysiology of OA, including increased expression of collagen type I, MMP-13, DDR2, and Runx-2. These effects include the loss of DDR1 expression [104]. In humans and mice, DDR2 knockout leads to severe craniofacial and skeletal malformations, such as changed dwarfism, cranial shape, diminished trabecular and cortical bone, and alveolar bone/periodontal deficiencies. In primary calvarial osteoblast cultures and mesenchymal cell lines, DDR2 knockdown/knockout inhibited osteoblast development, while DDR2 overexpression stimulated it. Reduced osteoprogenitor or osteoblast cell proliferation and differentiation may contribute to the poor bone repair associated with DDR2 deficiency [140]. 

Moreover, a crossbred of DDR1−/− and Ldle−/− mice develop atherosclerosis, in which the mice lack the DDR2 gene, long bones become shorter and flat bones become irregular. The CT evaluation confirmed it. DDR1 and DDR2 knockout mice have different skull development during live-CT imaging. At the same time, no research has been carried out to determine bone mineral density in DDR2 knockout mice. However, the loss of collagen binding in the DDR2 knockout mice delayed wound healing and lower tensile strength [141]. In addition, augmentation of DDRs and MMP-13 expression was observed in various in vivo animal studies induced by OA.

OA-induced genetic induction and surgically operated OA-induced models were utilized to explore the DDRs’ role in bone physiology. Besides, patients were found to have increased DDRs with MMP-13 expression in cartilage [142,143]. However, these findings contradicted a subsequent study by Holt et al. (2012), who discovered the same changes previously reported, such as DDR and MMP-13 upregulation in different mouse models and sedc mice. As a result, overexpression of DDRs is observed as one of the critical and early changes in many mouse models induced with OA, because researchers decided to reduce DDR expression in animal models generated with OA, facilitate cartilage degeneration leads to bone/joints speedy recovery for indirect proportional chances [144]. 

According to the previous study, when DDR2 and collagen II interact, they activate various downstream signaling pathways and dozens of effective terminator molecules, such as IL-15 and Dkk-1, which contribute to inflammation of cells entering membranes and causing cartilage and bone destruction in RA [118]. According to the US Food and Drug Administration, the most potent RTK inhibitor against severe bone damage developed arthritis in mice models is dasatinib. DDR2 inhibitors, on the other hand, are less effective due to their low binding specificity for collagen molecules [120]. DDR expression is lower in RA after OA, and there is little research on this disease. It is essential to remember that research in the field of DDRs is ongoing. As of our knowledge, no DDR-specific treatments have been licensed to treat OA or RA despite the promise of DDRs as therapeutic targets in these disorders. It is necessary to do additional research into the processes behind DDR signaling and their function in the pathogenesis of OA and RA to create effective DDR-targeted therapy.

The pericellular matrix (PCM) comprises several molecules interacting to generate a distinct extracellular matrix network. These components include collagen types VI and IX, biglycan, matrilin, fibronectin, and fibromodulin [145]. The PCM is frequently damaged or changed in osteoarthritis conditions. This can happen due to several physical and biological factors, including mechanical stress, inflammation, and OA-related biochemical changes. According to Cherry et al. (2020), pericellular matrix degradation may happen early in the progression of OA, causing abnormal chondrocyte–type II collagen interactions and initiating the catabolic signaling process. Type II collagen is the most abundant collagen in cartilage, which is the source of phosphorylate DDR-2 [146]. Due to its interaction with collagen type II, the DDR2 exhibits increased expression.

Additionally, Xu et al. (2011) use a mouse model with inducible DDR2 overexpression in cartilage to study the effects of the chondrocyte pericellular matrix on DDR2. The activation of DDR2 interacts with native collagen type II, which increases receptor expression in chondrocytes and induction of MMP13 expression [16]. The pericellular matrix typically surrounds articular cartilage and bone chondrocytes, whereas collagen II fibrils are found in territorial and interterritorial matrices, though they are not directly related to chondrocytes [147]. Although only surgery can jeopardize the medical meniscus, which hastens the development of OA by activating DDR, an independent study using chondrocyte cultures (cells without pericellular matrix) and in vitro, chondrocytes demonstrated the presence of collagen-induced MMP-13 stimulation (with organized pericellular matrix). All these findings suggest that DDRs may also play a role in RA pathogenesis [148]. 

## 8. Conclusions

It is widely accepted that DDR1 and DDR2 are pivotal in responding to numerous signaling pathways in bone formation, regeneration, as well as cartilage-related diseases. DDR1 and DDR2 expression dysregulation delay bone growth and bone regeneration, and targeting the DDR1 and DDR2 signaling pathways is an intriguing strategy for bone research. Several types of small-molecule inhibitors have been developed to regulate the expression of DDR1 and DDR2. Therefore, a better comprehension of the mechanism underlying the pathological intervention of DDR1 and DDR2 in bone diseases is essential for developing more specific and efficient therapeutic agents for treating bone-related disorders. 

## Figures and Tables

**Figure 1 ijms-24-14895-f001:**
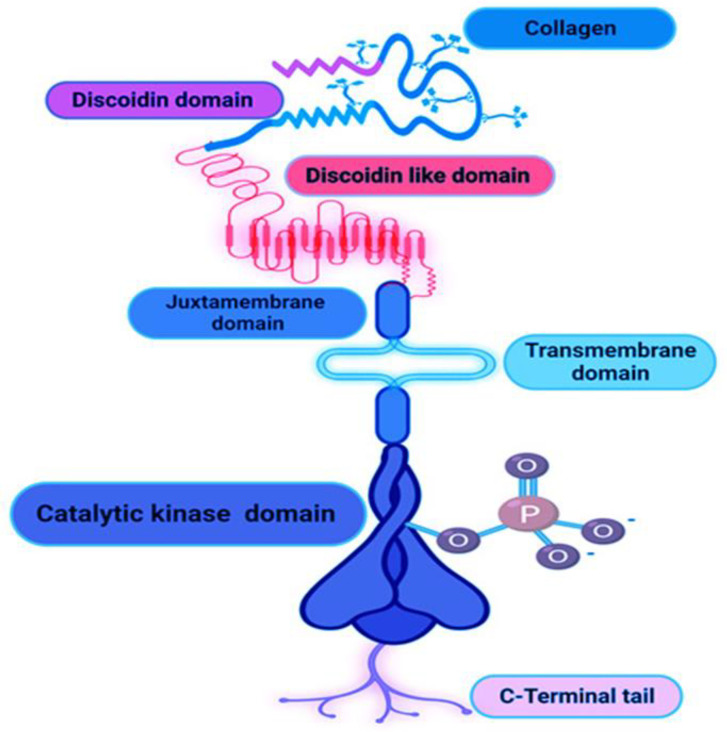
Structural diagram of DDR.

**Figure 2 ijms-24-14895-f002:**
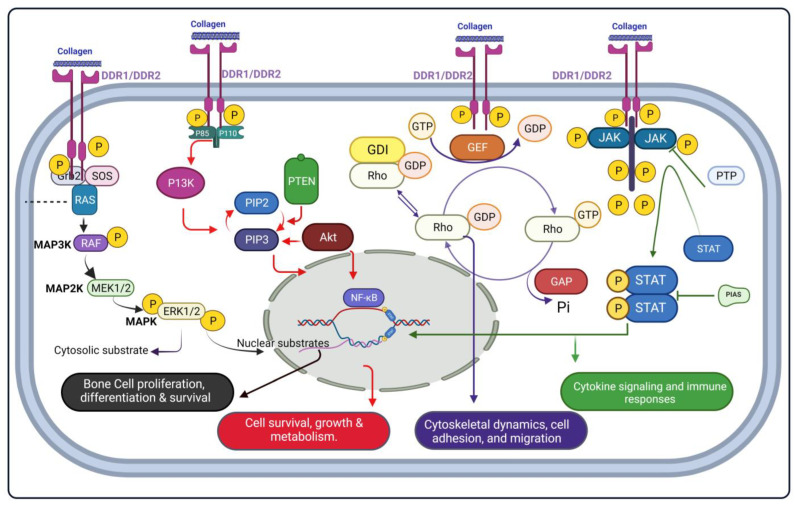
Schematic overview of possible DDR1/DDR2 associated signaling pathway {mito-gen-activated protein kinase (MAPK), Phosphoinositide 3-kinases (PI3Ks), Janus ki-nase/signal transducers and activators of transcription (JAK/STAT) and Rho-GTPase} in bone and cartilage. DDR1: Discoidin Domain Receptor Tyrosine Kinase 1; DDR2: Discoidin Domain Receptor Tyrosine Kinase 2; Grp-2: Growth factor receptor-bound pro-tein 2; Sos: Son of Sevenless; MPAK: Mitogen-activated protein kinase; MPAK 2: Mitogen-activated protein kinase 2; MPAK 3: Mitogen-activated protein kinase 3; ERK1:Extracellular signal-regulated kinase 1; P85: Phosphatidylinosi-tol-4,5-bisphosphate 3-kinase 85 kDa; P110: Phosphatidylinositol-4,5-bisphosphate 3-kinase 110 kDa; PIP2: Phosphatidylinositol 4,5-bisphosphate; PIP3: Phosphatidylino-sitol-3,4,5-triphosphate; PTEN: Phosphatase and tensin homolog; NF-κB: Nuclear factor kappa B; GTP: Guanosine-5′-triphosphate; GDP: Guanosine diphosphate; GEF: Guanine nucleotide exchange factor; Rho: Ras homologous; Pi: Inorganic phosphate; JAK: Janus kinase; PTP: Protein tyrosine phosphatases; STAT: Signal transducers and activators of transcription; STAT P: Phosphorylated signal transducers and activators of transcription.

**Table 1 ijms-24-14895-t001:** Some of the essential clinical significance of DDR associated with different types of diseases.

Receptor	Disease/Disease Model	Clinical Significance	Reference
DDRI	Chondrocyte-specific DDR1 knockout mice	Controls chondrocyte activity during endochondral ossification	[8]
DDRI	Osteogenesis/Osteoblast-specific knockout mice	DDR1 controls osteoblast/osteocyte autophagy	[9]
DDRI	Osteogenesis/osteoblast-specific DDR1 knockout (OKOΔDdr1) mice	Osteogenesis is controlled by p38 phosphorylation, which also down-regulates the osteogenesis markers	[10]
DDRI	DDR1 inhibition on osteoarthritis	Injecting 7 rh intraarterially (IA) decreased chondrocyte apoptosis and boosted autophagy.	[11]
DDR2	Osteoarthritis/Col9a1−/− mice	In the knee joints of Col9a1−/− mice, MMP-13 and DDR2 protein expression and the amount of type II collagen were degraded.	[12]
DDR2	Osteoarthritis/human	Increased fragments of type II collagen produced from MMP-13, DDR2, and MMP-13 were seen in cartilage	[13,14]
DDR2	Osteoarthritis/heterozygous sedc mouse	Expression of HtrA1, Mmp-13, and DDR2. Cartilage fissuring and erosion were observed	[15]
DDR2	Osteoarthritis/transgenic Mice	Expression of DDR2 was increased in knee joints, and DDR2 accelerated OA progression	[16]
DDR1	Atherosclerosis/DDR1-null SMC	Reduced expression of MMP2 and MMP9, decreased proliferative and migratory response	[17]
DDR2	Carotid injury/DDR2 wild-type and knockout mice	Reduced SMC proliferation, MMP synthesis, and ECM synthesis.	[18]
DDR1 and DDR2	atherosclerosis and lymphangioleiomyomatosis/ smooth muscle cells	Collagen expression is downregulated, while matrix metalloproteinase (MMP) is induced.	[19]
DDR1	Atherosclerosis/Ldlr−/− mice	Development of atherosclerotic plaque, promoting inflammation and fibrosis	[20]
DDR1	Atherogenesis/dr1+/+; ldlr−/− and DDR1−/−;Ldlr−/−	Macrophage infiltration and accumulation, decreased adhesion/chemotactic invasion of type IV collagen	[21]
DDR1	Chronic renal failure/DDR1-deficient mice	Blunting of glomerular fibrosis and inflammation and prevention of proteinuria	[22]
DDR1	Kidney fibrosis in Alport syndrome/DDR1 expression in Col4a3−/− mice	Improved kidney function and reduced inflammation and fibrosis	[23]
DDR1	Glomerulonephritis/DDR1−/− mice	Protected the crescentic glomerulonephritis	[24]
DDR1and DDR2	Bleomycin-induced lung fibrosis/mouse	inflammation and fibrosis	[25]
DDR2	Chronic liver injury/DDR2+/+ and DDR2−/− mice	Enhanced the gelatinolytic activity, HSC density, and collagen deposition.	[26]
DDR2	Alcoholic liver disease/rat	Silencing DDR2 prevent early stage alcoholic liver disease.	[27,28]
DDR1	Cancer/MCF7 HCT116 cell line	DDR1 activates the MAPK, Ras/Raf/ERK signaling	[29]
DDR1 and DDR2	Lung cancer/phosphoproteomic approach	Analyses of phosphotyrosine signaling profiles reveal novel ALK and ROS fusion proteins and oncogenic kinases, including EGFR and c-Met.	[30]
DDR1, DDR2	Non-small cell lung carcinoma.	DDR1 is overexpressed. Collagen types I, II, III, IV, V, VIII, and XI encourage altered expression of DDRs, which aid in the malignant progression of NSCLC.	[31,32,33]
DDR2	Breast cancer/human	Tumor cell invasion via collagen-I-rich extracellular matrices is assisted by maintaining the EMT phenotype, enhanced ERK2 activation, and phosphorylation of the transcription factor SNAIL1.	[34,35]

## Data Availability

Data are contained within the article.

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
