# Peer review of "Exploring the Cellular and Molecular Mechanism of Discoidin Domain Receptors (DDR1 and DDR2) in Bone Formation, Regeneration, and Its Associated Disease Conditions"

_ijms, 2023, doi:10.3390/ijms241914895_

Round 1
Reviewer 1 Report
1. In this paper, collagen is the main activator of DDRs signaling, and it is also suggested that it is doubtful whether 3-D made collagen gels can provide mechanical or external forces to induce DDR phosphorylation. However, this paper does not discuss whether simple mechanical stimuli, such as stretching, stretching, shear force, osmotic pressure, etc. have an impact on the activation of DDRs. If there is evidence in relevant literature, please make a supplement. If not, please explain.
2. In this paper, the molecular signaling pathways of DDR1 and DDR2 in bone and cartilage are described respectively. If a pathway figure is added, it can better show how DDRs play a key role in regulating downstream signaling molecules. The contents of 322-327 are somewhat repetitive. Please condense the contents to remove redundancy.
3. In this paper, “DDRs speed up the advancement of cartilage degeneration, whereas OA patients can not initiate these expressions because their pericellular matrix is damaged” is mentioned. Please explain the relationship between DDRs and PCM if the explanation is not clear.
4. In the statement corresponding to 98 of this paper, the case of Ddr2 is different from other statements in this paper, and should be changed to DDR2.
5. In the statement corresponding to 344 of this paper, EC is written incorrectly, so it should be changed to ECM.
6. In the statement corresponding to 367 of this paper, the closing parenthesis is missing.
7. In the corresponding sentences of 388 in this paper, the description of "sperm" in "female sperm" is wrong.
8. The language description of 492 corresponding statements in this paper is problematic.
9. In the statements corresponding to 494 in this paper, Postn and POSTN are case insensitive.
10. In this paper 498 corresponding statements, Ddr2, case error, should be changed to DDR2.
11.In the statement corresponding to 551 of this paper, Mmp13, case error, should be changed to MMP13; Cl10a1 is written incorrectly, should be changed to Col10a1.
12.In the sentences corresponding to 557 in this paper, the term "inhibitors" is not appropriate.
13.In the statement corresponding to 567 in this paper, the case of Ddr1 is different from other statements in this paper, so it should be changed to DDR1.
14.In the statement corresponding to 574 in this paper, the case of runx2 is different from other statements in the paper, so it should be changed to Runx2.
15.Statements corresponding to 599-608 in this paper repeatedly describe the theory that DDRs is triggered up-regulated in OA patients.
16.In this paper, there are repeated statements corresponding to 611-615, please delete them.
There are a number of grammatical and spelling errors that persist. Here is a short but definitely not complete list. These should be corrected, along with other grammatical and spelling issues.
Author Response
Response to Reviewers-1 comments- IJMS-2624859
Dear Editor's/Reviewer,
Thank you for giving me the opportunity to submit a revised draft of my manuscript titled “Exploring the cellular and molecular mechanism of discoidin domain receptors (DDR1 and DDR2) in bone formation, regeneration, and its associated disease conditions” to the International Journal of Molecular Sciences.
We appreciate the time and effort you and the reviewers have dedicated to providing valuable feedback on my manuscript. We are grateful to the reviewers for their insightful comments on my paper. We have been able to incorporate changes to reflect all the suggestions provided by the reviewers.
The grammatical and typical error is highly regretted. The typographical errors are removed, and the manuscript is now carefully corrected by a native English speaker; corrections have been made throughout the revised manuscript.
We have highlighted the changes within the manuscript. We hope the revised manuscript will fulfill the maximum criteria for your esteemed journal.
Here is a point-by-point response to the reviewers' comments and concerns.
Reviewer-1: Comments and Suggestions for Authors
- In this paper, collagen is the main activator of DDRs signaling, and it is also suggested that it is doubtful whether 3-D-made collagen gels can provide mechanical or external forces to induce DDR phosphorylation. However, this paper does not discuss whether simple mechanical stimuli, such as stretching, stretching, shear force, osmotic pressure, etc. impact the activation of DDRs. If there is evidence in relevant literature, please make a supplement. If not, please explain.
Response: Thanks for the suggestions. The suggested information were appropriately cited in the revised manuscript (Page No: 7 Line No: 260-263).
“Discoidin domain receptors (DDR1 and DDR2) have been identified as the receptor tyrosine kinases activated upon collagen binding. However, there is a lack of evidence regarding the effect of DDRs on the mechanical interaction between fibroblasts and ECM. ECM consists of various proteins, including collagen and proteoglycan. Lund et al. (2009) documented the interaction of DDR1 with three-dimensional type I collagen using a 3D culture of human mesenchymal stem cells. The dynamic cell shape changes and ECM microstructure tuning cause DDR1 and two-dimensional osteogenesis pathways to interact and modify their functions. Besides, the fibroblast dendritic extensions in 3D collagen matrices were entangled with matrix fibrils, causing integrin-independent mechanical interaction. However, the molecular pathways are unclear.
References
Kim D, You E, Min NY, Lee KH, Kim HK, Rhee S. Discoidin domain receptor 2 regulates the adhesion of fibroblasts to 3D collagen matrices. International Journal of Molecular Medicine. 2013;31(5):1113-8.
Jiang H, Grinnell F. Cell–matrix entanglement and mechanical anchorage of fibroblasts in three-dimensional collagen matrices. Molecular biology of the cell. 2005;16(11):5070-6.
Lund AW, Stegemann JP, Plopper GE. Mesenchymal stem cells sense three dimensional type I collagen through discoidin domain receptor 1. The Open Stem Cell Journal. 2009;1:40.
- In this paper, the molecular signaling pathways of DDR1 and DDR2 in bone and cartilage are described respectively. If a pathway figure is added, it can better show how DDRs play a key role in regulating downstream signaling molecules. The contents of 322-327 are somewhat repetitive. Please condense the contents to remove redundancy.
Response: Thanks for your valuable comments to improve the quality of the manuscript. As you suggested, the molecular signaling pathways of DDR1 and DDR2 in bone and cartilage are represented in the figure (Figure 2) with a description (Page No:11; Line No: 472-483).
Figure 2. Schematic overview of possible DDR1/DDR2 associated signaling pathway {mitogen-activated protein kinase (MAPK), Phosphoinositide 3-kinases (PI3Ks), Janus kinase/signal transducers and activators of transcription (JAK/STAT) and Rho-GTPase} in bone and cartilage
“Discoidin Domain Receptors (DDR1/2) are the members of Receptor Tyrosine Kinases (RTKs). There are two isoforms, DDR1 and DDR2. The phosphorylation of the intracellular tyrosine domain decides the faith of the signaling cascade. Collagen an extracellular matrix structural protein found in connective tissue, is the ligand for the activation of DDR1/2. Approximately 28 types of collagens are identified, but only a few are involved in activation. DDR1/2 has a crucial role in bone and cartilage development. The interactive pathways of DDR1/2 are MAPK, PI3K, JAK/STAT, and Rho-GTPase. The signal transduction induced by DDR1/2 also activates ERK1/2, Akt, cytokine signaling, and RhoA signaling pathway, respectively, and play a role in cytoskeletal dynamics, cell proliferation, differentiation, survival, adhesion, migration, cell metabolism, cytokine signaling, and immune responses. The possible molecular signaling pathways of DDR1 and DDR2 in association with bone and cartilage are shown in Figure 2”.
- In this paper, “DDRs speed up the advancement of cartilage degeneration, whereas OA patients can not initiate these expressions because their pericellular matrix is damaged” is mentioned. Please explain the relationship between DDRs and PCM if the explanation is not clear.
Response: According to your suggestion, the relationship between DDRs and PCM was included in the revised manuscript (Page No:15; Line No: 704-713).
“Pericellular matrix (PCM) comprises of unique extracellular matrix network. These include collagen types VI and IX, biglycan, matrilin, fibronectin, and fibromodulin (Alcaide-Ruggiero et al., 2023). PCM damage is common in osteoarthritis, and mechanical stress, inflammation, and OA-related metabolic alterations are the common causative agents. Cherry et al. (2020) proposed that early pericellular matrix degradation may cause aberrant chondrocyte–type II collagen connections and catabolic signaling in OA. Type II collagen is the most prevalent in cartilage, which produces phosphorylate DDR-2. Interaction with collagen type II increases DDR2 expression. In addition, Xu et al. (2011) examined the effects of the chondrocyte pericellular matrix on DDR2 in a mouse model with inducible DDR2 overexpression in cartilage DDR2 activation interacts with native collagen type II to promote chondrocyte receptor expression and Mmp13 gene expression”.
References
Alcaide-Ruggiero L, Cugat R, Domínguez JM. Proteoglycans in Articular Cartilage and Their Contribution to Chondral Injury and Repair Mechanisms. International Journal of Molecular Sciences. 2023;24(13):10824.
Chery DR, Han B, Li Q, Zhou Y, Heo SJ, Kwok B, Chandrasekaran P, Wang C, Qin L, Lu XL, Kong D. Early changes in cartilage pericellular matrix micromechanobiology portend the onset of post-traumatic osteoarthritis. Acta biomaterialia. 2020;111:267-78.
Xu L, Polur I, Servais JM, Hsieh S, Lee PL, Goldring MB, Li Y. Intact pericellular matrix of articular cartilage is required for unactivated discoidin domain receptor 2 in the mouse model. The American journal of pathology. 2011;179(3):1338-46
.
- In the statement corresponding to 98 of this paper, the case of Ddr2 is different from other statements in this paper, and should be changed to DDR2.
Response: The typical errors are highly regretted, and now we have fixed the typical mistake.
- In the statement corresponding to 344 of this paper, EC is written incorrectly, so it should be changed to ECM.
Response: The typical errors are highly regretted, and now we have fixed the typical mistake.
- In the statement corresponding to 367 of this paper, the closing parenthesis is missing.
Response: The typical errors are highly regretted, and now we have fixed the technical error.
- In the corresponding sentences of 388 in this paper, the description of "sperm" in "female sperm" is wrong.
Response: The typical errors are highly regretted, and now we have fixed the typical mistake.
- The language description of 492 corresponding statements in this paper is problematic.
Response: The grammatical error is highly regretted, and now we have fixed the typical mistake.
- In the statements corresponding to 494 in this paper, Postn and POSTN are case insensitive. Response: The typical errors are highly regretted, and now we have fixed the typical mistake.
- In this paper 498 corresponding statements, Ddr2, case error, should be changed to DDR2. Response: The typical errors are highly regretted, and now we have fixed the typical mistake.
- In the statement corresponding to 551 of this paper, Mmp13, case error, should be changed to MMP13; Cl10a1 is written incorrectly, should be changed to Col10a1.
Response: The typical errors are highly regretted, and now we have fixed the typical mistake.
- In the sentences corresponding to 557 in this paper, the term "inhibitors" is not appropriate.
Response: According to your comments, we include the appropriate word for ‘receptor’. Thank you very much.
- In the statement corresponding to 567 in this paper, the case of Ddr1 is different from other statements in this paper, so it should be changed to DDR1.
Response: The typical errors are highly regretted, and now we have fixed the typical mistake.
- In the statement corresponding to 574 in this paper, the case of runx2 is different from other statements in the paper, so it should be changed to Runx2.
Response: The typical errors are highly regretted, and now we have fixed the typical mistake.
- Statements corresponding to 599-608 in this paper repeatedly describe the theory that DDRs is triggered up-regulated in OA patients.
Response: Thanks for the suggestion. All sections have been revised accordingly.
- In this paper, there are repeated statements corresponding to 611-615, please delete them.
Response: Revised accordingly. We ensure that there are no repetitions.

Reviewer 2 Report
The presented review article entitled "Exploring the cellular and molecular mechanism of discoidin domain receptors (DDR1 and DDR2) in bone formation, regeneration, and its associated disease conditions" aims to summarize the structure, biological activity, and selectivity of the discoidin domain receptors ( DDR1 and DDR2) in terms of bone and cartilage regulation.
The review covered aspects of the structure, genome, expression, signaling, and more specifically the significance of DDRs for human pathophysiology. DDRs as potential therapeutic targets for bone and cartilage-related diseases are also discussed.
The need for additional research regarding DDRs' signaling and function in the pathogenesis of OA and RA was also reported.
My proposal to the authors is to draw up a scheme summarizing the signaling pathways and functionality of the DDRs, particularly in relation to bone and cartilage.
There is a repetition of some of the information in the text that needs to be cleared.
Some of the references need to be updated with the latest publications.
Author Response
Response to Reviewers-2 comments- IJMS-2624859
Dear Editor's/Reviewer,
Thank you for giving me the opportunity to submit a revised draft of my manuscript titled “Exploring the cellular and molecular mechanism of discoidin domain receptors (DDR1 and DDR2) in bone formation, regeneration, and its associated disease conditions” to the International Journal of Molecular Sciences.
We appreciate the time and effort you and the reviewers have dedicated to providing valuable feedback on my manuscript. We are grateful to the reviewers for their insightful comments on my paper. We have been able to incorporate changes to reflect all the suggestions provided by the reviewers.
The grammatical and typical error is highly regretted. The typographical errors are removed, and the manuscript is now carefully corrected by a native English speaker; corrections have been made throughout the revised manuscript.
We have highlighted the changes within the manuscript. We hope the revised manuscript will fulfill the maximum criteria for your esteemed journal.
Here is a point-by-point response to the reviewers' comments and concerns.
Reviewer-2: Comments and Suggestions for Authors
- The need for additional research regarding DDRs' signaling and function in the pathogenesis of OA and RA was also reported.
Response: According to your suggestion, additional research regarding DDRs' signaling and function in the pathogenesis of OA and RA was included in the revised manuscript (Page No:12; Line No: 527-534; Page No:13; Line No: 571-547; Page No:14; Line No: 581-587; Page No:15; Line No: 704-717).
“A study by Manning et al. (2006) offers experimental evidence that DDR2 may be an attractive target for developing disease-modifying OA medications. DDR2 was condi-tionally eliminated from the articular cartilage of Aggrecan-CreERT2 mice. The pro-gressive process of articular cartilage deterioration was significantly slowed in the knee joints of DDR2-deficient mice compared to their control mice. Damage to articu-lar cartilage in the knee joints of mice was related to elevated expression levels of DDR2 and matrix metalloproteinase. These findings imply that DDR2 may be an ap-propriate target for developing disease-modifying OA medicines”
“Mu and colleagues discovered that inhibiting DDR2 lowers inflammation and joint destruction via the H19-miR-103a-IL-15/Dkk-1 axis, where DDR2 plays a stimulatory function in the development of RA”
“Liu et al. (2023) conditionally deleted the DDR2 in myeloid lineage cells to generate cKO mice to investigate the role of DDR2 in myeloid lineage cells. They found that cKO mice exhibited more severe inflammation in collagen antibody-induced arthritis (CAIA) and high-fat diet (HFD)-induced obesity, indicating the protective role of DDR2 against inflammation. Mechanistically, DDR2 promotes macrophage repolar-ization are systematically implicated in OA's pathogenesis from the M1 to M2 pheno-type and protects the systemic inflammation”.
References
Manning LB, Li Y, Chickmagalur NS, Li X, Xu L. Discoidin domain receptor 2 as a potential therapeutic target for development of disease-modifying osteoarthritis drugs. The American journal of pathology. 2016 Nov 1;186(11):3000-10.
Liu Q, Wang X, Chen Y, Ma X, Kang X, He F, Feng D, Zhang Y. Ablation of myeloid discoidin domain receptor 2 exacerbates arthritis and high fat diet induced inflammation. Biochemical and Biophysical Research Communications. 2023 Mar 15;649:47-54.
Mu N, Gu JT, Huang TL, Liu NN, Chen H, Bu X, Zheng ZH, Jia B, Liu J, Wang BL, Wang YM. Blockade of discoidin domain receptor 2 as a strategy for reducing inflammation and joint destruction in rheumatoid arthritis via altered interleukin‐15 and dkk‐1 signaling in fibroblast‐like synoviocytes. Arthritis & Rheumatology. 2020 Jun;72(6):943-56.
- My proposal to the authors is to draw up a scheme summarizing the signaling pathways and functionality of the DDRs, particularly in relation to bone and cartilage.
Response: Thanks for your valuable comments to improve the quality of the manuscript. As you suggested, the molecular signaling pathways of DDR1 and DDR2 in bone and cartilage are represented in the figure (Figure 2) with a description (Page No:11; Line No: 472-483).
“Discoidin Domain Receptors (DDR1/2) are the members of Receptor Tyrosine Kinases (RTKs). There are two isoforms, DDR1 and DDR2. The phosphorylation of the intracellular tyrosine domain decides the faith of the signaling cascade. Collagen an extracellular matrix structural protein found in connective tissue, is the ligand for the activation of DDR1/2. Approximately 28 types of collagens are identified, but only a few are involved in activation. DDR1/2 has a crucial role in bone and cartilage development. The interactive pathways of DDR1/2 are MAPK, PI3K, JAK/STAT, and Rho-GTPase. The signal transduction induced by DDR1/2 also activates ERK1/2, Akt, cytokine signaling, and RhoA signaling pathway, respectively, and play a role in cytoskeletal dynamics, cell proliferation, differentiation, survival, adhesion, migration, cell metabolism, cytokine signaling, and immune responses. The possible molecular signaling pathways of DDR1 and DDR2 in association with bone and cartilage are shown in Figure 2”.
Figure 2. Schematic overview of possible DDR1/DDR2 associated signaling pathway {mitogen-activated protein kinase (MAPK), Phosphoinositide 3-kinases (PI3Ks), Janus kinase/signal transducers and activators of transcription (JAK/STAT) and Rho-GTPase} in bone and cartilage
- There is a repetition of some of the information in the text that needs to be cleared.
Response: Thanks for the suggestion. All sections have been revised accordingly.
- Some of the references need to be updated with the latest publications.
Response: There are some mistakes in citing old references, which had been revised. Thank you very much.
